# HOT PATE: PRIVATE AGGREGATION OF DISTRIBUTIONS FOR DIVERSE TASKS

## ABSTRACT

The Private Aggregation of Teacher Ensembles (PATE) framework Papernot et al. (2017) is a versatile approach to privacy-preserving machine learning. In PATE, teacher models are trained on distinct portions of sensitive data, and their predictions are privately aggregated to label new training examples for a student model. Until now, PATE has primarily been explored with classification-like tasks, where each example possesses a ground-truth label, and knowledge is transferred to the student by labeling public examples. Generative AI models, however, excel in open ended *diverse* tasks with multiple valid responses and scenarios that may not align with traditional labeled examples. Furthermore, the knowledge of models is often encapsulated in the response distribution itself and may be transferred from teachers to student in a more fluid way. We propose *hot PATE*, tailored for the diverse setting. In hot PATE, each teacher model produces a response distribution and the aggregation method must preserve both privacy and diversity of responses. We demonstrate, analytically and empirically, that hot PATE achieves privacy-utility tradeoffs that are comparable to, and in diverse settings, significantly surpass, the baseline "cold" PATE.

## 1 INTRODUCTION

Generative AI models, such as large language models (LLMs), are incredibly powerful tools that can be fine-tuned for specific contexts, even without explicit supervision Radford et al. (2019); Brown et al. (2020). Generative AI models diverge from conventional machine learning models in that they support open ended, *diverse* tasks, where there are multiple appropriate responses, and this very flexibility is essential for much of their functionality. Diversity is typically tuned via a temperature parameter in the softmax, with higher temperature yielding higher entropy (more diverse responses). Furthermore, when evaluating the coverage or extracting knowledge from a trained model, such as for distillation tasks, the conventional approach involves querying the model on a prepared (sampled or curated) test set of examples. However, with generative AI models, the knowledge coverage on a specific domain is often encapsulated by the output distribution itself to a general instruction as part of a *prompt* to the model, and can be evaluated or retrieved by sampling this distribution.

Frequently there is a need to train models or fine-tune publicly-available foundation models using sensitive data such as medical records, incident reports, or email messages. In this case, privacy must be preserved in the process. Specifically, we consider the strong mathematical guarantees of differential privacy (DP) Dwork et al. (2006); Dwork & Roth (2014). An approach that achieves privacy by modifying the training process is DPSGD Abadi et al. (2016), where noise is added to clipped gradient updates. DPSGD can also be applied with fine tuning Yu et al. (2022); Duan et al. (2023). An alternative approach to private learning, that only relies on black box training and use of models that are not privacy-preserving, is Private Aggregation of Teacher Ensembles (PATE) Papernot et al. (2017); Bassily et al. (2018); Papernot et al. (2018). PATE follows the "sample and aggregate" method (Nissim et al., 2007). We describe the basic workflow which we refer to here as *cold* PATE.

**The cold PATE framework**

1. The sensitive dataset $D$ of labeled training examples is partitioned into $n$ parts $D = D_1 \sqcup \cdots \sqcup D_n$. A *teacher* model $M_i$ is trained on data $D_i$ for $i \in [n]$.

2. Unlabeled examples are sampled from the public distribution. For each such example $x$ do as follows: For each teacher $i \in [n]$, apply $M_i$ to $x$ and obtain a label $y_i := M_i(x) \in V$. Compute the frequencies for $j \in V$

$$c_j = \sum_{i \in [n]} \mathbb{1}\{y_i = j\} \tag{1}$$

and privately aggregate $\boldsymbol{c}$ to obtain a single label $y \in V$ (or abort if there is insufficient agreement).

3. Use the newly labeled privacy-preserving labeled examples $(x, y)$ to train a student model.

The cold PATE workflow is limited by its formulation for classification-like tasks, where each example $x$ has a single ground-truth label $y \in V$, and the need for a source of unlabeled non-private training examples to facilitate the knowledge transfer to the student. Generative AI models support tasks with responses that are diverse and open ended. Moreover, knowledge is encapsulated in the diversity of the response distribution and there is a promise of transferring knowledge to the student in a more fluid way. We thus ask the following question:

> *Can we design a version of PATE that is effective for diverse and open-ended tasks and unleashes more of the capabilities of generative models?*

One motivation for our study is the effectiveness of in-context learning via *prompts*. A prompt is an engineered prefix with a task that is given to the base model. Prompts can include specific instructions and/or a set of *shots* (scenario examples). Prompts are appealing for multiple reasons: A small number of shots Liu et al. (2021) often outperform tailored trained models Zhou et al. (2022); Garg et al. (2023). Prompting is efficient, as it is simply inference – there is no need for parameter updates. Finally, prompts only requires API access to the model, which is important given the trend towards proprietary models.

When the data we have for the fine-tuning is sensitive, we would like the end product to be privacy-preserving. Concretely, consider generating a representative set of synthetic privacy-preserving data records from a set of sensitive data records. The sensitive records may include component that are identifying and components that are shared with many other records. A privacy-preserving aggregation ensures that the synthetic records do not include identifying information. We also need to preserve diversity in order to ensures coverage, that is, that our set of synthetic records is indeed representative. The synthetic records that are generated can then be used to train a student model that is not necessarily generative. Or they can be used to construct student prompts that are privacy preserving for downstream tasks. The latter allows for harnessing the ability of generative models to generalize from few examples.

Concretely, we seek a PATE mechanism that supports the following. Each teacher is assigned a disjoint subset of sensitive data records. These data records are used to construct a prompt that also includes an instruction of the form "generate a representative data record given this examples set of data records." Each teacher then has its own distribution on responses. By repeating multiple times we can obtain different samples that are a representative set of shots. We then hope to aggregate responses of different teachers in a way that preserves both diversity and privacy.

A benefit of using prompts is that there is little cost to scaling up the number of teachers – each teacher is simply a prompted base model and there is no need for training or significant storage. The bottleneck to scaling up is therefore the amount of available sensitive data. Scaling up the number of teachers is highly beneficial because generally with DP aggregation, the number of queries we can support for a given privacy budget grows quadratically with the number of teachers.

**Overview** In this work we propose *hot PATE*, described in Section 2. The framework is suitable for auto-regressive models and diverse and open ended tasks, where the appropriate response is a sample from a distribution. With hot PATE, each teacher $i \in [n]$ at each step computes a distribution $\boldsymbol{p}^{(i)}$ over tokens $V$. These distributions are aggregated so that the response token from the ensemble is sampled from that aggregate distribution. The aggregation method should preserve privacy but critically, to ensure knowledge transfer, should also preserve the diversity of the teachers distributions. Our primary technical contribution is formalizing this requirement and designing aggregation methods with good privacy utility tradeoffs.

In Section 3 we motivate and formalize a definition of *preserving diversity* that allows for knowledge transfer while being mindful of the limitations imposed by privacy. Informally, for a parameter $\tau \in [n]0$, we require that any token that has probability at least $q > 0$ (no matter how small) across $c$ teachers where $c \geq \tau$, is "transferred" in that it has probability $\Omega(qc/n)$ in the aggregate distribution. We also require that we do not transfer irrelevant tokens, that is, for any token $j$, its probability in the aggregate distribution is not much higher than its average probability in the teacher distributions. We then demonstrate that a natural approach for diversity-preserving aggregation, where each teacher contributes a token $y_i$ sampled independently from $\boldsymbol{p}^{(i)}$, inherently exhibit a poor privacy-utility tradeoff, where utility deteriorates with the diversity of teacher distributions: When $q$ is small enough, even tokens with broad support $c \geq n/2$ can not be transferred.

In Section 4 we propose *ensemble coordination*, which is the primary ingredient for designing a privacy-preserving aggregation method where utility does not decrease with diversity. The coordinated ensemble samples a shared randomness and based on that, each teacher $i$ contributes a token $y_i$. The marginal distribution of each $y_i$ is $\boldsymbol{p}^{(i)}$, same as with independent samples. But the key difference is that teachers votes are highly positively correlated. This means that the frequency $c_j$ of token $j$ has high spread and in particular can (roughly) be $\Omega(\tau)$ with probability $\Omega(q)$. This property is the key for achieving DP aggregation with no penalty for diversity. In Section 5 we empirically demonstrate the properties and benefits of ensemble coordination using a simple example on the GPT3.5 interface.

In Section 6 we propose DP aggregation schemes that preserve diversity when applied to frequency histograms generated by coordinated ensembles. We distinguish between applications with *homogeneous* or *heterogeneous* ensembles. The underlying assumption with homogeneous teachers, same as with cold PATE, is that most teachers have the core knowledge we wish to transfer. In this case, diversity preservation with $\tau > n/2$ suffices. Heterogeneous teachers correspond to a setting where each teacher is an agent of one or few users. In this case, we want to preserve diversity both within and between teachers and allow smaller groups of teachers to support each prediction, that is, use a smaller $\tau$. We explore, analytically and empirically, data-dependent privacy analysis and demonstrate potential for order of magnitude gains over DP composition in the number of queries.

**Related work** The recent work of Duan et al. (2023) adapted PATE to working with prompts: Each part $D_i$ of the data was used to create a text prompt $T_i$. The ensemble is then used to label curated queries. But while some design elements were tailored to LLMs, the workflow and privacy analysis were identical to cold PATE Papernot et al. (2018) and inherited its limitations. The original submission proposing PATE Papernot et al. (2017) included a discussion (Appendix B.1) of using more of the teachers histogram than the maximizer for distillation tasks. They concluded that it is beneficial for utility but does not justify the privacy loss. Despite the superficial resemblance, this is very different from what we do. The token sampled from the aggregate distribution is in a sense also the (noisy) maximizer of teacher agreement. The subtlety is that this token is still a sample – we "force" the teachers to agree but there is a distribution on the agreement token. Finally, there is a very rich literature on PATE extensions that go beyond classification tasks. The works we are aware of address different problems and use different techniques than hot PATE. For example, PATE had been used for image generation using generative adversarial networks (GAN). In Jordon et al. (2018), a student discriminator is trained using teacher discriminators and a cold-PATE like labeling approach. In Long et al. (2021), a student generator is trained by aggregating the gradients produced by teachers discriminators, with private aggregation of the gradient vectors. The technical component is the private aggregation of the gradients and is a different problem in a different context than hot PATE.

## 2 HOT PATE

We use the term *tokens* for elements of the input and response strings and denote the vocabulary of tokens by $V$. For an input context (prompt), the response sequence is generated sequentially token by token. For diverse tasks, tokens are sampled from a probability distribution over $V$. The probabilities are computed from weights $(w_j)_{j \in V}$ computed by the model and a *temperature* parameter $t > 0$, using a softmax function:

$$p_j := \frac{e^{w_j/t}}{\sum_{i \in V} e^{w_i/t}} .$$

In low temperatures, the highest weight token $\arg\max_j w_j$ has probability close to 1. As we increase the temperature, the probability distribution flattens with similarly-weighted tokens having similar probabilities. *Cold* temperature is appropriate for classification-like tasks with one correct response and *hot* temperature is appropriate for diverse tasks. We therefore refer to the outlined PATE workflow as cold PATE and to our proposed workflow as hot PATE.

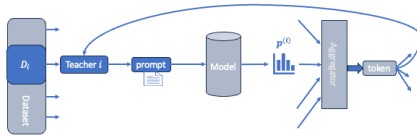

Figure 1: Hot PATE with an auto-regressive base model

Hot PATE (see illustration in Figure 1) partitions $D$ to disjoint parts $D_i$ ($i \in [n]$) and constructs a prompt $T_i$ from data part $D_i$. We then generate a sanitized response sequence $R$ of tokens. We initialize $R \leftarrow \{\}$ and proceed sequentially in lockstep, by repeating the following:

1. For $i \in [n]$: Let $\boldsymbol{p}^{(i)}$ be the output distribution over $V$ when querying the model with the prompt $T_i$<instruction to complete prefix>$R$.
2. Apply a DP and diversity-preserving randomized aggregation $\mathcal{M}((\boldsymbol{p}^{(i)})_{i \in [n]}) \mapsto y$, where $y \in V$.
3. Concatenate $R \leftarrow R \parallel y$.

This design is open-ended and assumes that the instructions are effective in producing students prompts or components for such prompts, such as representative shots. This assumption aligns with the demonstrated and evolving capabilities of contemporary large language models, as well as the progress made in prompt engineering. An underlying requirement with both cold and hot PATE is that a sufficient number of teachers possess the knowledge we wish to transfer. In both cases the ensemble's purpose is to privately transfer that knowledge to the student. The key distinction is that with cold PATE, knowledge coverage is achieved by sampling examples from the input distribution (and then labeling them by the ensemble). In hot PATE, the intent is that coverage is attained organically, through the broad range of diverse responses generated in response to a general instruction within the prompt. The requirement of preserving diversity, that we will make more precise in the sequel, is needed in order to facilitate this knowledge transfer. We would like the aggregate distribution, the output distribution of $\mathcal{M}((\boldsymbol{p}^{(i)})_{i \in [n]})$, to retain the diversity of individual teacher distributions $(\boldsymbol{p}^{(i)})_{i \in [n]}$.

## 3 PRIVATE AND DIVERSE AGGREGATION

Diversity and privacy appear to be conflicting in that DP requires that the output token is supported by sufficiently many teachers, a "reporting threshold" that depends on the privacy parameter values. But preserving diversity means that tokens with low probability also need to be transferred to the student.

The gold standard for preserving diversity is the average teacher distribution $\frac{1}{n}\sum_{i \in [n]} \boldsymbol{p}^{(i)}$. But this is not privacy preserving because tokens that have positive probabilities with only one or few teachers are identifying and should not be released. Fortunately, we can settle for a weaker notion of preserving diversity that is more robust. The premise in PATE is that the patterns of interest are captured by many or even most teachers. Therefore, low probability across many teachers is something we care to transfer whereas high probability in few teachers, the "bad case" for privacy (and robustness), may not be something we have to transfer. The average distribution does not distinguish the two cases, so it can not be a starting point. We first formalize our nuanced diversity preservation notion:

**Definition 1** (Diversity-preserving aggregation of distributions). Let $f(\boldsymbol{p}^{(i)})_{i \in [n]} \mapsto \boldsymbol{P}$ map from $n$ probability distributions over $V$ to a probability distribution over $V \cup \{\perp\}$. We say that $f$ is *diversity-preserving* with $\tau \in \mathbb{N}$, $\beta \in (0, 1]$, $\gamma \geq 1$ if for any input and $j \in V$

1. For all $q \in [0, 1]$,

$$(c_{j,q} := \sum_{i \in n} \mathbb{1}\{p_j^{(i)} \geq q\}) \geq \tau \implies P_j \geq \beta \cdot \frac{c_{j,q}}{n} q \,.$$

2. $P_j \leq \gamma \frac{1}{n} \sum_{i \in [n]} p_j^{(i)}$

The first requirement is that probability $q$ across enough ($\tau$) teachers, no matter how small is $q$, is transferred to the aggregate distribution. The second ensures that we do not output irrelevant tokens.

Requirements are stricter (and can be harder to satisfy) when $\beta$ and $\gamma$ are closer to 1 and when $\tau$ is smaller. A setting of $\tau = 0$ and $\beta = \gamma = 1$ allows only for the average distribution to be the aggregate. A larger $\tau$ increases robustness in that more teachers must support the transfer.

**Remark 1** (failures). *We allow $\perp$ (failure) in the support of the aggregate distribution because under the DP requirement there are input distributions (for example, those with disjoint supports, e.g. responses to instructions that ask for a patient ID) where no token can be returned. Hot PATE has several options to work with failure responses: (i) The step can be repeated (different shared randomness may yield a token), (ii) a response token can instead be sampled from a non-private default prompt or model, or (iii) the prompt instructions can be redesigned.*

**Remark 2** (Setting of $\tau$). *In homogeneous ensembles, most teachers receive a representative part of the data and possess the knowledge we wish to transfer. This occurs when we use a random partition so that most teachers obtain a representative set of data records. In this case, we aim to transfer the parts of the distributions that are common to most teachers and $\tau > n/2$ suffices. In heterogeneous ensembles, each teacher might have data from one or very few "users." This arises when each teacher has small capacity (prompts currently have limited size of 8k-64k tokens OpenAI (2023b)) or when by design each teacher is an agent of a single user. In this situation, we aim to transfer parts of the distribution that are common to smaller subgroups of teachers and set $\tau \ll n$, possibly as low as permitted under the privacy requirement.*

Before describing DP aggregation methods that satisfy Definition 1, we instructively examine a scheme that can not satisfy the requirements, as it exhibits an inherent privacy-diversity tradeoff: Sample independently $y_i \sim \boldsymbol{p}^{(i)}$ for each teacher $i \in [n]$, compute frequencies $c_j$ as in (1), and apply *any* DP aggregation to the histogram $\{(j, c_j)\}$ (as with cold-PATE). Now consider the case of identical teacher distributions that are uniform over $k$ special tokens with probability $q = 1/k$ each. From Definition 1, each of the $k$ special tokens needs to be reported with probability at least $\beta/k$. But the frequencies $c_j$ of these tokens are concentrated around $c_j \approx n/k$. In terms of DP, each frequency value $c_j$ has sensitivity 1 and for large enough $k$, the counts drop below the "DP reporting threshold" of our privacy parameters and therefore none of these tokens can be reported. To transfer these distributions through such a frequencies histogram we need to adjust the DP parameters to allow for reporting threshold to be below $n/k$, that is, to decrease proportionally to $k$. Therefore, *any* DP aggregation of this histogram can not satisfy Definition 1 in that it would fail for a sufficiently large $k$. We run into the same issue if we define our histogram with $c_j := \sum_i p_j^{(i)}$ (as proposed in Duan et al. (2023)). The issue again is that the maximum frequency decreases with diversity ($k$).

The approach where each teacher contributes a sample, however, is appealing as it "factors out" the distributions: Instead of aggregating distributions, we work with a histogram of frequencies. But with independent sampling we arrived at a dead end – and it may seem that we need to ditch the sampling approach all together. Fortunately, our proposed aggregation method also samples teacher distributions to generate a histogram of frequencies. The difference is that the frequency of a token is not concentrated around its expectation. A tokens $j$ that broadly has a low probability $q$ will appear, sometimes, with very high frequency $c_j$ that does not depend on $q$. What does depend on $q$ is the probability of this event. This allows it to pass through a high "privacy threshold."

## 4 ENSEMBLE COORDINATION

*Ensemble coordination*, described in Algorithm 1, is a randomized mapping from a set of $n$ probability distributions over $V$ to a histogram over $V$ with total count $n$. We sample shared randomness $\rho$. For each teacher $i \in [n]$ we compute $y_i \in V$ that is a function of $\rho$ and $\boldsymbol{p}^{(i)}$. We then compute the frequencies $c_j$ for $j \in V$, as in (1), and return the frequency histogram.

---

**Algorithm 1:** `CoordinatedSamples`

---

**Input:** Teacher distributions $(\boldsymbol{p}^{(i)})_{i \in [n]}$

**foreach** *token* $j \in V$ **do** draw iid $u_j \sim \mathsf{Exp}[1]$     // Draw shared randomness $\rho = (u_j)_{j \in V}$

**foreach** *teacher* $i$ **do**                               // Draw coordinated samples $(y_i)_{i \in [n]}$

   $y_i \leftarrow \arg\max_j \frac{p_j^{(i)}}{u_j}$                       // bottom-$k$ sampling transform

**foreach** *token* $j \in V$ **do**                                    // Compute frequencies

   $c_j \leftarrow \sum_{i \in [n]} \mathbb{1}\{y_i = j\}$

**return** $\{(j, c_j)\}_{j \in V}, \rho = (u_j)_j$// Histogram of frequencies

---

Importantly, ensemble coordination over prompts can be implemented via an enhanced API access to the model. The best approach is to support the shared randomness $\rho$ as input along with the query. Alternatively, we can use API access that returns the distribution over tokens – The current OpenAI text completion interface returns the five highest probabilities OpenAI (2023b).

The sampling method in ensemble coordination is a classic technique called *coordinated sampling*. The technique was first introduced in statistics applications in order to obtain samples that are stable under distribution shifts Kish & Scott (1971); Brewer et al. (1972); Saavedra (1995); Rosén (1997); Ohlsson (2000). It was then introduced in computer science for sampling-based sketches and a form of Locality Sensitive Hashing (LSH) Cohen (1994; 1997); Broder (2000); Indyk & Motwani (1998).

Similarly to independent sampling, the marginal distribution of $y_i$ for each teacher $i$ is simply $\boldsymbol{p}^{(i)}$. Therefore, the expected frequency of token $j$ is

$$\mathsf{E}_\rho[c_j] = \sum_i p_j^{(i)} \,. \tag{2}$$

The key difference is that votes of different teachers are highly positively correlated. For two teacher distributions $i, k$, the probability of them having the same sample is the weighted Jaccard similarity of the distributions:

$$\Pr_\rho[y_i = y_k] = \frac{\sum_j \min\{p_j^{(i)}, p_j^{(k)}\}}{\sum_j \max\{p_j^{(i)}, p_j^{(k)}\}}$$

In particular, when two distributions are identical, the samples are the same $y_i = y_k$.

We establish that the respective requirements of Definition 1, diversity-transfer and relevance, can be satisfied by only selecting tokens that appear with high frequency in the histogram. We show that a token $j$ for which $m$ teachers $i$ have $p_j^{(i)} > q$ has frequency at least $m/2$ with probability at least $0.34q$ (see proof in Appendix A):

**Lemma 1** (diversity transfer). *For any token $j$ and $p, q \in [0, 1]$,*

$$\Pr_\rho\left[c_j \geq p \cdot \sum_{i \in n} \mathbb{1}\{p_j^{(i)} \geq q\}\right] \geq \frac{1}{2} \ln(1/p)q$$

To establish relevance we show that high frequency must have a "backing." The following is immediate from (2) and Markov's inequality (and is tight in the sense that for any $T$ there are distributions where equality holds):

**Lemma 2** (relevance). *For any token $j$ and $T$,*

$$\Pr_\rho[c_j \geq T] \leq \frac{1}{T} \sum_{i \in [n]} p_j^{(i)}$$

Therefore, broadly speaking, it is possible to satisfy the requirements of Definition 1 by reporting only tokens with frequency that is $\Omega(\tau)$, where $\tau$ is the required teachers support. Our DP aggregation methods are presented in Section 6.

## 5 EMPIRICAL DEMONSTRATION

We demonstrate the properties of coordinated ensembles using the OpenAI GPT3.5 text completion interface OpenAI (2023b). Given a text prompt, the interface provides the tokens and probabilities of the top-5 tokens. We generated $10^3$ queries (prompts) of the following form (see Example in Figure 2) and collected the top-5 tokens and their probabilities.

```
On planet Z, some numbers are edible.      <name>   from
planet Z eats the following numbers for breakfast:
<random permutation of {63, 56, 28, 17} ∪ {m ∼ U{11,...,99}}> Give me
an example breakfast number in planet Z. Respond with
just the number.
```

The top 5 tokens returned in all of the $10^3$ queries were 2 digit decimal numbers. The response token was more likely to be one of the example numbers in the prompt than a different number.

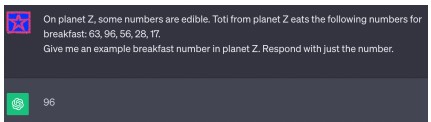

Figure 2: Query to GPT3.5

Our queries were constructed to have a shared "general" component that we aim to capture via the private aggregation: The four common numbers that we color-code in plots 17,28,56, 63. Other components such as the name and the fifth number are considered "private." A limitation of the interface is that we can not obtain the full distribution over tokens. We thus scaled up each partial distribution of top-5 to obtain a distribution $\boldsymbol{p}^{(i)}$ for queries $i \in [10^3]$.

Figure 3 (left) reports the distribution of the average probabilities $10^{-3} \sum_{i=1}^{10^3} \boldsymbol{p}^{(i)}$ of each token with a positive probability. The model displayed some preference for 63 over the three other special numbers. The right plot is a histogram of the frequencies (normalized by $10^3$) obtained by independently sampling one token $y_i$ from each distribution $\boldsymbol{p}^{(i)}$. There was little notable change between different sampling: For each token $j$, the frequency is a sum of independent Poisson random variables with parameters $p_j^{(i)}$, that we know from standard tail bounds to be concentrated around its expectation.

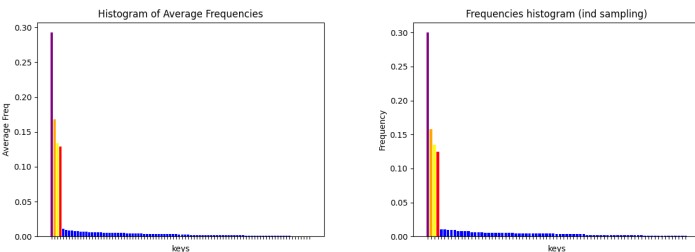

Figure 3: Average probabilities (left) and normalized frequency histogram from independent samples (right)

Figure 4 reports example frequency histograms obtained with coordinated sampling (Algorithm 1) for three samples of the shared randomness $\rho$. Note that a different special token dominates each histogram, and the maximum frequency is much higher than the respective expected value.

Figure 5 reports aggregate results for $10^3$ frequency histograms produced for each of coordinated and independent samples. From each histogram we collected the highest and second highest frequencies of a special number and the highest frequency of a non-special number. The left plot shows the counts (sorted in decreasing order) of each of these three values. Note that with independent samples, frequencies remain close to their expectations: The top frequency corresponds to that

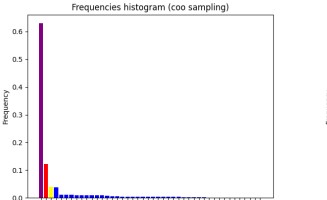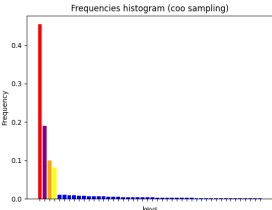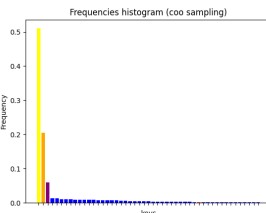

Figure 4: Histograms from coordinated samples for different sampling of shared randomness $\rho$

of 63. The second highest to one of the other special numbers. Note that with independent sampling no token (special or not) in no trial had frequency $> 0.5$. Moreover, the gap between the top and second frequencies was consistent and reflected the gap of the expected frequencies between the two top special tokens.

With coordinated samples, about half of the trials had a dominant token with frequency $> 0.5$. The dominant token was always one of the special tokens, but not necessarily the special token with the highest average frequency. Figure 5 (right) shows the probability of each of the special numbers to have frequency above $> 0.5$. We can see that all four special numbers are represented with probability roughly proportional to their average probability.

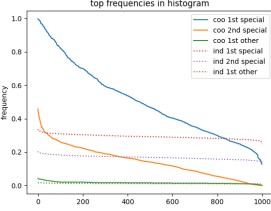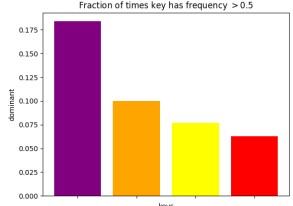

Figure 5: Counts of top frequencies in decreasing order (left). Distribution of dominant token (right)

We observe two benefits of coordinated sampling. First, tokens appear with high frequency, which is easier to report privately. Second, when there is dominance, there tends to be a large gap between the highest and second highest frequencies, which is beneficial with data-dependent privacy analysis.

Due to the limitation of the interface that returns only the top 5 probabilities, we constructed our example to have $k = 4$ special tokens that should be transferred to the student distribution. Note that the benefits of coordinated sampling scale up with $k$: With $k$ special tokens, the top frequency with independent sampling decreases proportionally to $k$ whereas the top frequency with coordinated sampling remains high and does not depend on $k$. With larger $k$, the lines for coordinated sampling in Figure 5 (left) would remain the same whereas the lines for independent sampling would shift down proportionally to $k$.

## 6 Aggregation Methods of Frequency Histograms

Our aggregation methods are applied to frequency histograms generated by a coordinated ensemble and return a token or $\bot$. We propose two meta schemes that preserves diversity in the sense of Definition 1: One for homogeneous ensembles, where we use $\tau > n/2$, in Section 6.1 and one for heterogeneous ensembles, where $\tau \ll n/2$ (but large enough to allow for DP aggregation), in Section 6.2. We then discuss DP implementations that admit data-dependent privacy analysis. The latter allows for many more queries for the same privacy budget: The privacy loss does not depend on queries with no yield, with high agreement, or with agreement with a public prior. With heterogeneous ensembles we can also gain from individualized per-teacher privacy charging. For privacy analysis, it suffices to consider the histogram in isolation, as it has the same sensitivity as vote histograms with cold PATE: When one teacher distribution changes, one token can gain a vote

and one token can lose a vote. This because the shared randomness $\rho$ is considered "public" data. Diversity preservation is considered for the end-to-end process from the teacher distributions.

## 6.1 HOMOGENEOUS ENSEMBLES

---
**Algorithm 2:** `DistAgg` homogeneous

---
$\boldsymbol{c}, \rho \leftarrow$ `CoordinatedSamples`$((\boldsymbol{p}^{(i)})_{i \in [n]})$         // Algorithm 1
$(j, \hat{c}_j) \leftarrow$ `NoisyArgMax`$_L(\boldsymbol{c})$       // DP noisy maximizer with error $L$
**if** $\hat{c}_j > (n/2 + L)$ **then return** $j$ **else return** $\perp$

---

When $\tau > n/2$, there can be at most one token $j$ with frequency $c_j \geq \tau$. If there is such a token, we aim to report it. Otherwise, we return $\perp$. Our scheme is described in Algorithm 2 in terms of a noisy maximizer (`NoisyArgMax`$_L$) procedure. The latter is a well studied construct in differential privacy McSherry & Talwar (2007); Durfee & Rogers (2019); Qiao et al. (2021). Generally, methods vary with the choice of noise distribution and there is a (high probability) additive error bound $L$ that depends on the privacy parameters and in some cases also on the support size and confidence. For our purposes, we abstract this as `NoisyArgMax`$_L$ that is applied to a frequency histogram $\boldsymbol{c}$ and returns $(j, \hat{c}_j)$ such that $|c_j - \hat{c}_j| < L$ and $\max_{h \in V} c_h - c_j \leq 2L$. We show that the method is diversity preserving (proof is provided in Appendix A):

**Lemma 3** (Diversity-preservation of Algorithm 2). *For $\mu > 1$, Algorithm 2, instantiated with* `NoisyArgMax`$_L$ *as described, is diversity preserving in the sense of Definition 1 with $\tau = \mu(n/2 + 2L)$, $\beta = \ln(\mu)/2$ and $\gamma = 2$.*

The two most common noise distributions for DP are Gaussian and Laplace noise. (Cold) PATE was studied with both. The Gaussian-noise based Confident-GNMax aggregator Papernot et al. (2018); Duan et al. (2023) empirically outperformed the Laplace-based LNMAX Papernot et al. (2017) on cold PATE. for Algorithm 2. The advantages of Gaussian noise are concentration (less noise to separate a maximizer from low frequency tokens), efficient composition, and more effective data dependent privacy analysis. Laplace-based noise on the other hand can preserve sparsity (a consideration as the key space of tokens or strings of token can be quite large), there is an optimized mechanism with sampling (for medium agreement), and there are recent improvement on data-dependent privacy analysis across many queries (the situation with hot PATE) Cohen & Lyu (2023). Our privacy analysis in Section 7 uses a data-dependent Laplace-based approach.

## 6.2 HETEROGENEOUS ENSEMBLES

---
**Algorithm 3:** `DistAgg` Heterogeneous

---
$\boldsymbol{c}, \rho \leftarrow$ `CoordinatedSamples`$((\boldsymbol{p}^{(i)})_{i \in [n]})$        // Algorithm 1
Sample $j \in V$ with probability $\frac{c_j}{n}$     // Weighted sampling of a token from $\boldsymbol{c}$
**if** $c_j \geq 2L$ **then return** $j$ **else return** $j$ *or* $\perp$

---

For lower values of $\tau$, we propose the meta-scheme described in Algorithm 3: We perform weighted sampling of a token from $\boldsymbol{c}$ and return it if its count exceeds $2L$. If it is below $2L$ we may return either $j$ or $\perp$. We propose DP implementations in Section 8. We establish that Algorithm 3 is diversity-preserving (proof provided in Appendix A).

**Lemma 4** (Diversity-preservation of Algorithm 3). *For $\mu > 1$, Algorithm 3 is diversity preserving in the sense of Definition 1 with $\tau = \mu 2L$, $\beta = \frac{1}{2\mu} \ln(\mu)$ and $\gamma = 1$.*

## CONCLUSION

We proposed and evaluated hot PATE, an extension of the PATE framework, that facilitates open ended private learning via prompts. The design is based on a notion of robust and diversity-preserving aggregation of distributions that can be implemented in a privacy preserving way. We expect our design to have further applications.

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

## 7 PRIVACY ANALYSIS CONSIDERATIONS

Hot PATE issues a large number of queries for token by token generation. The privacy loss can be mitigated by scaling up the number of teachers $n$. Prompts are relatively inexpensive. The current OpenAI API supports $10^5$ context/output tokens for \$5-\$10 OpenAI (2023a). Therefore the limiting factor on $n$ is the amount of available private data for creating homogeneous teachers (see Remark 2) rather than training cost or model storage. Scaling up $n$ is important as (due to composition theorems for differential privacy) the number of queries we can issue to the ensemble increases *quadratically* with the number $n$ of teachers.

Additionally, we demonstrate empirically and analytically significant gains from data dependent privacy analysis. Broadly speaking, privacy loss is higher on "borderline" queries where the aggregation has two or more likely outputs. That is, queries that return a particular token with high probability or return $\perp$ with high probability incur very little privacy loss. We demonstrate that when conditioning on the sampled shared randomness, only a small fraction of frequency histograms are "borderline." Moreover, for queries with low *yield* (large probability of $\perp$ response and low probability of returning a token), the total privacy loss only depends on yield responses.

For queries we used Algorithm 2, where for `NoisyArgMax` we used Cohen et al. (2021) with the maximum sanitized frequency, with privacy parameters $(\varepsilon_0, \delta_0)$. To analyse privacy across queries

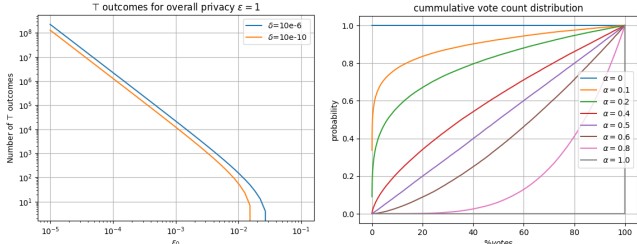

Figure 6: Left: Number of $\top$ responses for $\varepsilon_0$-DP queries for total $\varepsilon = 1$ loss. Right: Cummulative maximum frequency for varying common part $\alpha$.

in a data-dependent way we applied the boundary-wrapper method in the Target Charging Technique (TCT) of Cohen & Lyu (2023). The wrapper modifies slightly the output distribution of the query algorithm (conditioned on $\rho$) to include an additional outcome $\top$ (*target*). The probability of $\top$ increases for "borderline" queries and is at most $1/3$. The technique allows us to analyse the privacy loss by only counting target hits. Figure 6 (left) reports the number of $\top$ (target) responses we can have with the boundary wrapper as a function of $\varepsilon_0$ with overall privacy budget is $\varepsilon = 1$. When $\varepsilon_0 \leq 0.01$, it is about $(10\varepsilon_0)^{-2}$.

We consider teacher distributions with probability vectors of the form $\boldsymbol{p}^{(i)} = \alpha \cdot \boldsymbol{s} + (1 - \alpha) \cdot \boldsymbol{r}^{(i)}$, where $\boldsymbol{s}$ and $\boldsymbol{r}^{(i)}$ are probability vectors. That is, with probability $\alpha$ there is a sample from the common distribution $\boldsymbol{s}$, and with probability $(1-\alpha)$, there is a sample from an arbitrary distribution. The distribution of the maximum frequency $c$ of a token is dominated by sampling $y \sim \mathsf{Exp}[\alpha]$ and then $c \sim \mathsf{Bin}[e^{-y \cdot (1-\alpha)}, n]$. It is visualized in Figure 6 (right) for varying values of $\alpha$. Qualitatively across all $\alpha > 0$, there is probability $\approx \alpha$ of being above a high threshold (and returning a token).

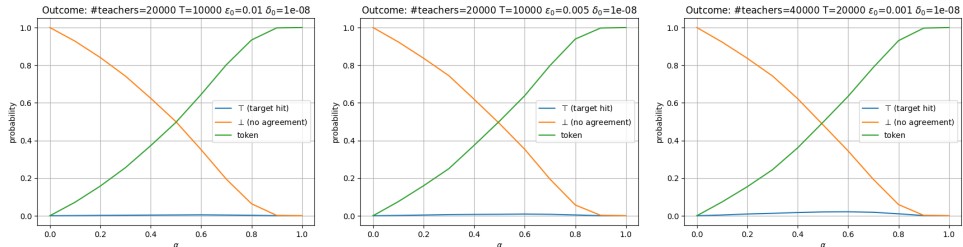

Figure 7: Sweep of $\alpha$, showing probabilities of outcomes: token, $\bot$, $\top$ (target hit).

Figure 7 shows the distribution of responses as we sweep $\alpha$, broken down by $\top$ (target hit), $\bot$ (abort), and token (yield). The number of queries per target hit, which is the inverse of the probability of $\top$, is $\gtrapprox \varepsilon_0 n$. It is lowest at $\alpha \approx T/n$ and is very high for small and large $\alpha$, meaning that the privacy cost per query is very small.

The yield (probability of returning a token) per query is $\approx \alpha$. Note that as $\alpha$ decreases, both yield and target probabilities decrease but their ratio remains the same: In the regime $\alpha \leq T/n$, the yield per target hit is $\approx \varepsilon_0 n/2$. Queries with $\alpha \gg T/n$ are essentially free in that the yield (token) probability is very high and the $\top$ (target hit) probability is very low.

When using $n = C_\delta/\varepsilon_0$ ($C_\delta \approx 2\log(1/\delta_0)$) and plugging this in, we obtain that we get $\gtrapprox 0.005\frac{1}{C_\delta}n^2$ yields for overall privacy budget $\varepsilon = 1$. This means that we pay only for yield and not for queries. This holds in the "worst case" across all $\alpha$ values, but the number of yields can be much higher when queries have large $\alpha$ (and "yields" do not incur privacy loss).

## 8  DP METHODS FOR HETEROGENEOUS ENSEMBLES

We propose two DP methods to implement Algorithm 3 (Section 6.2) with different trade offs. In both cases we can apply data-dependent privacy analysis so that queries that do not yield a token (that is, return $\bot$) are essentially "free" in terms of the privacy loss. The parameter $L$ depends on the privacy parameters (and logarithmically on $|V|$).

**Private Weighted Sampling**  This methods gains from sparsity but the calculation of privacy loss is for the whole ensemble. We can do the analysis in the TCT framework Cohen & Lyu (2023) so that privacy loss only depends on yield queries (those that return a token). We perform weighted sampling by frequency of each token to obtain the sampled histogram $c'$ and then sanitize the frequencies of sampled tokens using the end-to-end sparsity-preserving method of Cohen et al. (2021) to obtain $c^*$. The sanitizing prunes out some tokens from $c'$ with probability that depends on the frequency $c_j$, privacy parameters, and sampling rate. All tokens in $c'$ with frequency above $2L$, where $L$ only depends on the privacy parameters, remain in $c^*$.[1] The final step is to return a token from $c^*$ selected uniformly at random or to return $\bot$ if $c^*$ is empty. We mention the related earlier (non optimized) sparsity-preserving methods Bun et al. (2019); Korolova et al. (2009); Vadhan (2017) and optimized but not sparsity-preserving Ghosh et al. (2012).

**Individual Privacy Charging**  This method does not exploit sparsity, but benefits from individual privacy charging Kaplan et al. (2021); Cohen & Lyu (2023). It is appropriate when $2L \ll n$. The queries are formulated as counting queries over the set of teachers. The algorithm maintain a per-teacher count of the number of counting queries it "impacted." A teacher is removed from the ensemble when this limit is reached. Our queries are formed such that at most $O(2L)$ teachers (instead of the whole ensemble) can get "charged" for each query that yields a token.

To express Algorithm 3 via counting queries we do as follows: We draw a sampling rate of teachers uniformly from $1/n, \dots, 1$ and then uniformly draw a token from $v \in V$. We sample the teachers with this sampling rate and count the number $c'_v$ of sampled teachers with $y_i = v$. We then do a `BetweenThresholds` test on $c'_j$ (using Cohen & Lyu (2023) which improves over Bun et al. (2017)) to check if $c'_v \geq 2L$. For "above" or "between" outcomes we report $v$. If it is a "between" outcome we increment the loss counter of all sampled teachers with $y_i = v$ (about $2L$ of them). We note that the actual implementation is much more efficiently and does not require this "blind" search.

Teachers that reach their charge limit get removed from the ensemble. The uniform sampling of the sampling rate and token emulated weighted sampling. The probability that a token gets selected is proportional to its frequency. The sub-sampling of teachers ensures that we only charge the sampled teachers. Teachers are charged only when the query is at the "between" regime so (with high probability) at most $\approx 2L$ teachers are charged. Because we don't benefit from sparsity, there is overhead factor of $\log(|V|(n/L))$ in the privacy parameter (to contain the error in so many queries) but we gain a factor of $n/L$ by not charging the full ensemble for each query in the case where most teachers have different "solutions" to contribute.

## A  PROOFS

*Proof of Lemma 1.* Let $i$ be such that $p_j^{(i)} \geq q$. Fix the sampled min value $x \sim \mathsf{Exp}[q]$ for $q$ part of the probability of $j$. We get that

$$\Pr[y_i = j] \geq \Pr_{y \sim \mathsf{Exp}[1-q]}[y > x] = e^{-x(1-q)}$$

For $x < \frac{-\ln p}{1-q}$ we have that the probability is at least $e^{-x(1-q)} \geq p$. Different teachers that share part of the distribution can only be positive ly correlated. So we get that if there are $c_{j,q}$ teachers with $p_j^{(i)} \geq q$ then the distribution of the number of teachers with $y_i = j$ dominates $\mathsf{Bin}[e^{-x(1-q)}, c_{j,q}]$,

---

[1]We remark that the method also produces sanitized (noised) frequency values $c_j^*$ for tokens in $c^*$ such that $|c_j^* - c_j| \leq L$. And hence can also be used for `NoisyArgMax`

which for any $x \leq \frac{-\ln p}{1-q}$ dominates $\mathsf{Bin}[p, c_{j,q}]$. So with probability at least $1/2$, we have at least $pc_{j,q}$ teachers with $y_i = j$.

This happens with probability at least

$$\Pr_{x \sim \mathsf{Exp}[q]}[x < \frac{-\ln p}{1-q}] = 1 - e^{(\ln p)q/(1-q)} \geq -(\ln p)q$$

For $p = 1/2$ we get that the probability is $\geq 0.69q$. For $p = 2/3$ it is $\geq 0.4q$.

$\square$

*Proof of Lemma 3.* We apply Lemma 1 with $p = 1/\mu$. We obtain that the token $j$ has frequency at least $c_j \geq n/2 + 2L$ with probability at least $0.5 \ln(\mu)q$. Therefore we have $\hat{c}_j \geq n/2 + L$ with probability at least $0.5 \ln(\mu)q$. Note that a token can only be reported if its frequency is $c_j > n/2$. Using $T = n/2$ in Lemma 2 we obtain that the relevance requirement is satisfied with $\gamma = 2$. $\square$

*Proof of Lemma 4.* Consider the first requirement of Definition 1. Consider a token $j$ with $c_{j,q} \geq \tau$. From Lemma 1 using $p = 1/\mu$ we obtain that the token $j$ has frequency at least $c_j \geq c_{j,q}/\mu \geq 2L$ with probability at least $0.5 \ln(\mu)q$. The token is sampled with probability $\min\{1, kc_j/n\}$ and if so appears also in $\boldsymbol{c}^*$ (since $c_j \geq 2L$). The expected size (number of entries) of $\boldsymbol{c}^*$ is at most $k$ and thus it is returned if sampled with probability at least $1/k$. Overall it is sampled and reported with probability at least $\min\{1/k, c_j/n\}$. In total, the probability is $P_j \geq \min\{1/k, c_{j,q}/(\mu n)\}0.5 \ln(\mu)q \geq \frac{1}{2k\mu} \ln(\mu)\frac{c_{j,q}}{n}q$.

The second requirement of Definition 1 is immediate. The expected frequency of token $j$ is $\sum_{i \in [n]} p_j^{(i)}$ and it is sampled with probability at most $\frac{k}{n} \sum_{i \in [n]} p_j^{(i)}$. It can only be the output if sampled. $\square$

