# OpenReview forum: "Hot PATE: Private Aggregation of Distributions for Diverse Tasks"
_ICLR.cc/2024/Conference — Submitted to ICLR 2024_

### Official Review · Reviewer_mnAY · 2023-10-31

**Soundness:** 1 poor
**Presentation:** 1 poor
**Contribution:** 1 poor
**Rating:** 3
**Confidence:** 3

**Summary:**

The work proposes Hot PATE. It is the first time for me that it is really difficult to directly point out what the main contributions of this work are. The work claims that the standard PATE does not consider the diversity of responses among the teachers. In the proposed Hot PATE, the knowledge of the teacher is assumed to be captured by the diversity of responses. This approach proposes to use a temperature
parameter in the softmax to control the diversity of responses. The work is showcased on discrete prompts.

**Strengths:**

N/A

**Weaknesses:**

1. The paper builds directly on the recent work by Duan et al. 2023 [1] and proposes a very limited extension.
Moreover, it incorrectly states the method from the previous work [1]. Namely, this submission states that [1] "it requires a source of unlabeled non-private training examples to facilitate the knowledge transfer to the student". It is incorrect, [1] in Table 2 studies two settings, (IID Transfer) when the public dataset is from the same and (OOD Transfer) different distribution than the private training data. Moreover, [1] selects which tokens are taken into account as labels and is flexible in the selection of these tokens for the labels.
2. The main contribution is not related to the prompts. "The aggregation method should preserve privacy but to facilitate the knowledge transfer from teachers to the student, should critically also preserve the diversity of the teacher distributions. Our primary technical challenge was to formalize this requirement and design an aggregation method with a good privacy utility tradeoff." If this is the main contribution the hot PATE can be used for any task, not only for the discrete prompts.
3. It is not true that PATE was used only for the classification tasks. The submission claims: "Until now, PATE has primarily been explored with classification-like tasks, where each example possesses a ground-truth label, and knowledge is transferred to the student by labeling random examples." Many follow-ups for PATE consider, e.g., image generation: G-PATE https://proceedings.neurips.cc/paper/2021/hash/171ae1bbb81475eb96287dd78565b38b-Abstract.html "G-PATE: Scalable Differentially Private Data Generator via Private Aggregation of Teacher Discriminators" [2]

Additional comments:
- "utility must deteriorates with the diversity of teacher distributions." -> must deteriorate



** References: **
[1] Haonan Duan, Adam Dziedzic, Nicolas Papernot, and Franziska Boenisch. "Flocks of stochastic parrots: Differentially private prompt learning for large language models." NeurIPS 2023.
[2] Yunhui Long, Boxin Wang, Zhuolin Yang, Bhavya Kailkhura, Aston Zhang, Carl Gunter, Bo Li. "G-PATE: Scalable Differentially Private Data Generator via Private Aggregation of Teacher Discriminators." NeurIPS 2021.

**Questions:**

As in the weaknesses.

---

> ### Author Response · Authors · 2023-11-11
> **Response to review mnAY**
>
> Thank you for the review and the references!  Indeed, it seems that unfortunately there is confusion about understanding our contribution. We attempt to clarify by addressing the listed "weaknesses".
>
> W1. We do not "build directly" on Duan et al 2023.  Our technique addresses what we view as a significant deficiency in the design of Duan et al -- not effectively working with model diversity. Our technique is general, but we found the use case of learning via prompts (which Duan et al considered) very compelling.
>
> The core of our contribution is formalizing a notion of a diversity and privacy-preserving aggregate distribution.  We give a method where privacy loss does not increase with diversity that is critically important in diverse settings.
>
>  Our description of Duan et al is indeed simplified. We focused on what we deem relevant to make the points. The point is that the Duan et al approach did use unlabeled training examples for knowledge transfer. The reviewer points out that a non iid method was also used. But this detail is not very relevant to us. The point is that the mechanism for knowledge transfer is through unlabeled examples whereas with our approach, the diversity (and its preservation) allows for compelling alternatives (simply transfer the distribution). As for selecting a token among multiple: In Duan et al it is still the case that each teacher provides a vote for a token and these votes are privately aggregated. With increased diversity, there will not be a dominant vote and we incur higher privacy loss to identify a lower maximum.  This is exactly the issue our method corrects. Privacy loss does not increase with diversity!
>
> W2. Indeed the main technical contribution is a method for diversity-preserving private aggregation of distributions.  We view learning via promps as a very compelling use case. This use case motivated our design.
>
> W3. We simplified to focus on what is relevant. Indeed, "PATE" is broadly used for any sample-and-aggregate approach to DP machine learning and there are many works with this term.  The reviewer pointed to a GAN method called G-PATE.  This is not relevant to the way our approach extends on the original PATE or prompt-PATE.  G-PATE can be viewed as a different "branch" in the sample-and-aggregate private machine learning literature-- use private aggregation of gradients from teacher "discriminators" to train a student "generator." G-PATE and the related works do not address the issue we were out to address. We will explain this in a "related work" paragraph in the revised version.

---

> ### Comment · Reviewer_mnAY · 2023-11-11
> **Answer**
>
> Thank you for your quick response. In general, I still think that this paper requires a substantial amount of work and is definitely not ready to be accepted. The main problems are:
>
> 1. Poor writing and the lack of statement about what the main contribution is.
> 2. The experimental results should compare to the PATE method in general and not only to the PromptPATE from Duan et al. (Duan et al. addressed privacy protection from all types of prompts using both PATE and DP-SGD).
> 3. The scalability of the proposed solution may prove impractical for tasks of greater complexity.
>
> - W1. "Our technique addresses what we view as a significant deficiency in the design of Duan et al -- not effectively working with model diversity." - Duan et al. do not consider models but model adaptations.
> - W2. This needs to be clearly stated and then the relevant experimental design is required. The current one is too narrow.
> - W3. In my review, I pointed out the lack of sufficient related work in this submission and incorrect statements.
>
> I also read through the other reviews and fully agree with them. I maintain my score.

---

> ### Author Response · Authors · 2023-11-21
> **Revision**
>
> A revised version is now uploaded.  We thank the reviewer for the comments and tried to address to the extent we could what seemed to lead to some misunderstandings by the reviewer.  We also included a "related work" section that better explains the relation to Duane et al (which did not address the core problem that we address) and includes other references and relation to our work.
>
> Regarding the points made:   Problem (1) (lack of statement of what the main contribution is) is very subjective.  Other reviewers seemed to understand it at a high level.  We did try to improve this based on all comments in the revision.
>
> We disagree with problem (3) "the scalability of the proposed solution...."  This is in fact the whole point of coordinated ensembles.  That we do not "pay" for increased diversity.   It is established analytically (proofs) and there is really no actual need for an experimental design.  Experiments are needed in the event when there is hidden complexity in the practical setting that is not captured in the abstraction.  In this case, the analysis makes the point well. The small demonstration (that had limited diversity due to limitations of the API) shows the gains over a "naive" approach.  The naive approach is exactly what you would get if you apply Duane et al for the aggregation step. There is sharp drop in utility with diversity.
>
> We disagree with problem (2).  We do not think a comparison with DP-SGD fine tuning is relevant here.  It is true that Duane et al also considered DP-SGD fine-tuning (that is, not using prompts). We are not touching that space.  We only consider PATE-like settings with prompts as a motivation. We disagree with the characterization that it is "too narrow".   In-context learning via prompts is very promising and effective and we feel it is important to understand how to "do it right" with privacy.   Our approach is the way to do it.  It address the major specific technical hurdle.  Orthogonal issues are domain-specific prompt engineering and potential light-wait enhancements to the API (which we propose).

---

### Official Review · Reviewer_1aLU · 2023-10-31

**Soundness:** 2 fair
**Presentation:** 1 poor
**Contribution:** 2 fair
**Rating:** 3
**Confidence:** 3

**Summary:**

This paper proposes a modification/extension to the PATE framework, termed "hot PATE". Rather than relying solely on the noisy argmax, as in the traditional PATE method, hot PATE aggregates the distributions of teacher outputs. The key assertion is that by aggregating distributions, the diversity of responses is better preserved, which should enhance learning. Experiments have been conducted within the prompt learning paradigm

**Strengths:**

- The idea of increasing the information of the teacher aggregation beyond just the argmax seems both valid and intuitive.

**Weaknesses:**

- This submission reads more like a draft and doesn't seem ready for review. I found challenges in understanding some parts due to the clarity and quality of the writing.  Although I've attempted to interpret the content sentence by sentence, a significant portion of the text remains confusing and ambiguous (see examples below). In its current state, I believe the submission needs more work to meet the (writing) standards expected for an ICLR paper.

- The scope and contribution of this submission seems not clearly stated. Initially, the paper appears to claim a general extension of the PATE framework, but the method and experiment sections mainly focus on prompt learning. A clearer definition of the scope is needed for an accurate evaluation of the submission. Furthermore, since the distillation formulation (which aggregates distributions as this submission intends to do) was briefly discussed in the original PATE paper [1] (Appendix B.1), it's crucial to articulate the novelty and insights of this work in comparison to that prior research.

[1] "Semi-supervised knowledge transfer for deep learning from private training data", ICLR 2017

**Questions:**

- Recommendations for improvement include a possible rewriting of the work to enhance its readability. Specifically, it might be helpful to check the consistent and proper use of hyphens and to verify the correct application of \citet and \citep. Some sentences could benefit from rephrasing for clarity. Below are a few examples from the submission that need further clarification:
  - Diversity and privacy appear to be conflicting in that DP requires that the output token is supported
by sufficiently many teachers, a “reporting threshold” that depends on the privacy parameter values.
  - Therefore, low probability across many teachers is something we care to transfer whereas high
probability in few teachers, the “bad case” for privacy, is also not something we need to transfer.
  - A tokens j that broadly has a low probability q will appear, sometimes, with very high
frequency cj that does not depend on q. What does depend on q is the probability of this event. This allows it to pass through a high “privacy threshold.”

- Some "minor" points regarding Definition 1:
  - Missing left-bracket for $f(p^{(i)})_{i\in[n]})$
  - The notation $j\in V$ might not be rigorous. Perhaps $j$ should denote the *index* for words in $V$.
  - Why is the dependence on $\tau$ completely not reflected in the example shown on page 4

---

> ### Author Response · Authors · 2023-11-11
> **Response to review**
>
> We appreciate the feedback and interest. Our revised version has improved readability. Here we respond specifically to the "weaknesses" (which is where the questions are):
>
> The primary application of our privacy and diversity-preserving aggregation method is the Hot PATE setting.  The case when teachers have output distributions arise naturally with prompts and token generation so this is a very compelling application. Hot PATE broadly applies when (1) models have an output distribution and (2) it is important to transfer the diversity (e.g. the diversity is not "noise" on a ground truth but the signal itself). It does not have to be prompts.
>
> Thank you for pointing out B.1 in [1].
> Relation to our work:  It is not directly relevant except for the general message that transferring diversity is helpful for student learning (in the setting of distillation).  What B.1 in [1] say is that with cold PATE, transferring more of the teacher vote histogram (distribution) rather than just the argmax helps with accuracy. They also say there is additional privacy loss of reporting more and it may not be worth the gain.
> What we consider is the case where each teacher has a distribution and it is valuable to transfer this diversity to the student. An application of this setting is generation by LLMs.  Limiting LLMs to say argmax takes away from their value.  Our technical contribution is a method that preserves the diversity without incurring privacy loss over the cold PATE baseline.
>
> The "questions" seem more about writing style and readability and we will address them in the revision.  Thank you!

---

> > ### Author Response · Authors · 2023-11-21
> > **Revised version**
> >
> > A revised version is uploaded.  The scope is now better conveyed in the introduction. There is a related work paragraph.  There is an additional discussion on parameters (including $\tau$ that the reviewer commented on).

---

> > > ### Comment · Reviewer_1aLU · 2023-11-23
> > >
> > > Dear Authors,
> > >
> > > Thank you for your rebuttal. Unfortunately, I have noticed that the current version of the submission does not reflect significant improvements, as many of the ambiguous sentences I pointed out remain unchanged. Additionally, the incorrect usage of \citet and \citep persists. Consequently, my score for this submission will remain unchanged. In summary, I believe that a comprehensive rewrite and another round of the reviewing process would be necessary to properly assess the value of the paper.

---

### Official Review · Reviewer_JMJ6 · 2023-11-05

**Soundness:** 2 fair
**Presentation:** 1 poor
**Contribution:** 2 fair
**Rating:** 5
**Confidence:** 3

**Summary:**

This paper proposes the "hot PATE" which is an extension of the Private Aggregation of Teacher Ensembles (PATE) framework. Traditional PATE is primarily used for classification tasks with definitive ground-truth labels, which suffers limitations when applied to more open-ended, diverse tasks characteristic of generative AI models like large language models (LLMs). Hot PATE aims to address these challenges by enabling each teacher model to contribute a response distribution to preserve both privacy and the diversity of responses. This paper evaluates the hot PATE by conducting empirical demonstrations using the OpenAI GPT3.5 interface.

**Strengths:**

1. This paper presents an innovative extension of the PATE framework, enabling privacy-preserving learning in generative AI tasks.
2. This paper provides a thorough theoretical analysis of the hot PATE framework.

**Weaknesses:**

1. This paper is not well-written and is difficult to follow.
2. This paper does not include empirical validation of the student model's performance.
3. This paper does not mention related work, leaving it unclear whether there are other baseline works for comparison.
4. This paper does not provide sufficient detail about the methodologies used, especially regarding the implementation of the hot PATE framework and the experimental setup.

**Questions:**

1. What is w_j and how is it computed? Could the authors make this clearer?
2. Why are the experiments limited to only OpenAI GPT-3.5, and therefore, to only the top 5 tokens?
3. What is considered as private information in this paper?
4. What does the student model represent in the hot PATE framework?

---

> ### Author Response · Authors · 2023-11-11
>
> We appreciate the feedback and interest.
>
> Response to "weaknesses":
>
> W2 [empirical validation of student model]  We propose a meta scheme and demonstrate by running the numbers that large privacy-preserving students prompts can be generated effectively. Large scale applications would require orthogonal components such as datasets and tasks, prompt engineering, and more. This is beyond the scope of this work. Our work can be viewed as proposing a method that SHOULD be used IF applications arise in which PATE is applied with diverse tasks. This is established with theory and simulations. The method itself might have other applications.
>
>
> W3 We cited the related work we are aware of that seemed most relevant. There is significant number of works on PATE, Prompts, LLMs, out there.  If the reviewer is aware of relevant works that we had missed we would appreciate sharing that.
>
> W4 The revised version would include additional details.
>
> Response to questions:
>
> Q1: The notation w_j are the model weights that are transformed to probabilities using softmax.  This is very standard and the purpose was just to explain the process in which diversity is introducted and tuned in models. Our method just assumes there is an output distribution but applies regardless of how it is computed.
> Q2:The demo is indeed limited by the interface of OpenAI GPY 3.5 which is beyond our control.  It only provides the top 5 probabilities.  We would have larger gain over baseline with more diversity, and this is demonstrated in the simulations. We used this interface because this is what is available. Regardless, the purpose of the demo was to demonstrate that the models are indeed diverse.  Importantly, we included simulations in Section 7 that cover much more general problem parameters.
>
> Q3: The private information is the dataset that is used to generate the teacher prompts.  The dataset can be in the form of examples (e.g. medical records) that can be partitioned to teachers.
>
> Q4: The student model is a new prompt that is privacy preserving.  It can be a set of example "shots" or have another form.

---

### Official Review · Reviewer_5rq4 · 2023-11-06

**Soundness:** 3 good
**Presentation:** 2 fair
**Contribution:** 2 fair
**Rating:** 3
**Confidence:** 4

**Summary:**

The paper presents an extension of PATE, called HOT PATE, designed for generative AI models and tasks that require "diverse" responses.
The authors adapt the original PATE methodology to the generative model domain, where multiple valid responses exist, and differential privacy must be preserved.

**Strengths:**

1. The paper introduces an original concept by extending PATE to generative models.
2. I appreciated the discussion around the conflict between diversity and privacy.
3. The simplicity of the approach is a plus.

**Weaknesses:**

1. A critical weakness is the limited scope of the experimental evaluation. The evaluation is reported on a single experiments with only 5 outputs. This might not adequately reflect the framework’s performance with truly open-ended queries.
2. The proposed solution’s scalability might be impractical for more complex (and realistic) tasks.
3. The generalizability of the empirical results is questionable due to the specificity of the demonstration.
4. The paper lacks a clear guideline for balancing privacy trade-offs in various contexts.

**Questions:**

1. Can the authors elaborate on the framework's expected performance with fully open-ended queries and the plans for more comprehensive empirical evaluations?
2. What measures are in place to ensure the quality of outputs in more complex (and hopefully actual open-ended) scenarios, and how is output quality measured?
3. Can you also comment on the framework scalability and its impact on privacy?

---

> ### Author Response · Authors · 2023-11-11
>
> We appreciate the feedback and interest!  We agree that the experiments were fairly limited scope but we believe that the theory, together with simulations and the (limited) experiments, demonstrated the value of the method. We expect that our method is the way to apply PATE with diverse models and any effective method in future should use this approach. The method for private diversity-preserving aggregation may have additional applications as well.
>
> W1, W2, W3
> The demo was indeed limited by the GPT3.5 interface which only returns the top-5 probabilities. The purpose of the demo was meant to show that real models behave in the way we expect and when tasked with diverse tasks, indeed have diverse responses. Importantly, we demonstrated through the theory and also simulations that the more diverse the distributions are, the larger the gain from our approach over the prior naive methods of dealing with diversity.  Moreover, with our method there is no utility loss with diversity.  Utility only depends on the extent of agreement between teacher distributions, not on the "entropy" of the agreement components. This follows from the theory and demonstrated with simulations (Section 7). So as long as you believe the premise that models tend to cover the diversity of the answer space to a query, you need to see the value of the method.
>
>
> W2, W3
> The use case of prompts is interesting in that it allows for a very large number of teachers, when we have enough data,compared with traditional PATE (where each teacher is a trained model). We included a numerical analysis of privacy cost per yield (token) in some regimes and number of yields dependence on number of teachers. The number of yield queries increase quadratically with the number of teachers. For example, if we have 1M sensitive examples, we get 10 examples for each of 100k teachers. This can allow for large (10^5+ student prompts). Indeed, we do not have a concrete evaluation on a particular task and particular dataset, but we did run the numbers and they demonstrate the feasability.
>
> W4 we will expand on this (discuss regimes and guidelines) in the revised version.
>
> Response to questions:
>
> Q1 [Performance on fully open-ended queries] This submission is a proof of concept. We propose a method, and demonstrate its potential with theory and simulations and running the numbers. Actual implementation requires orthogonal components including datasets and prompt engineering that might be tailored to an application. When we have access to a large model beyond public interfaces or when publicly available interfaces provide more of the token distribution we can extend the experiments. As said, the beauty of our method is that there is no cost for diversity. This follows from the theory.
>
> Q2 [Quality of output] The revised version (will be uploaded soon) will include more discussion to that end to address questions we envision might come up in some applications.
>
> Q3 [Scalability/privacy] See Section 7 and response to W2,W3. We did run the numbers. As for impact on privacy. We use DP, with mathematical guarantees and run the numbers with overall epsilon<1. We have in mind a use case of highly sensitive data.

---

> > ### Comment · Reviewer_5rq4 · 2023-11-12
> > **Thanks**
> >
> > Thank you for your rebuttal.
> > As you state above, your solution is a proof of concept, and thus  I suggest to take into account mine and other reviewers comments prior a possible resubmission to make this into a work that lives up to its full potential.

---

### Meta-Review · Area_Chair_tacR · 2023-12-11

**Metareview:**

The paper presents an extension of the well-established PATE framework to generative models. While the idea is in principle new, the reviewers raised several concerns. First, empirical evaluation is limited, as experiments are reported for only a single dataset and a small number of output tokens. Secondly, the reviewers find the paper hard to follow. Third, there are insufficient details on the implementation of the framework and the experimental setup. Finally, the authors should consider articulating prior work on private generative models, and whether the are relevant baselines to compare against.

**Justification For Why Not Higher Score:**

The proposed application of the PATE framework for generative models does not seem to be the right approach.

**Justification For Why Not Lower Score:**

N/A

---

### Decision · Program_Chairs · 2024-01-16

Reject